# Anatomical Factors of the Anterior and Posterior Maxilla Affecting Immediate Implant Placement Based on Cone Beam Computed Tomography Analysis: A Narrative Review

**DOI:** 10.3390/diagnostics14151697

**Published:** 2024-08-05

**Authors:** Milica Vasiljevic, Dragica Selakovic, Gvozden Rosic, Momir Stevanovic, Jovana Milanovic, Aleksandra Arnaut, Pavle Milanovic

**Affiliations:** 1Department of Dentistry, Faculty of Medical Sciences, University of Kragujevac, 34000 Kragujevac, Serbia; 2Department of Physiology, Faculty of Medical Sciences, University of Kragujevac, 34000 Kragujevac, Serbia

**Keywords:** morphometric analysis, anterior maxilla, posterior maxilla, immediate implant placement, cone beam computed tomography (CBCT)

## Abstract

Background: The aim of this narrative review was to provide insights into the influence of the morphological characteristics of the anatomical structures of the upper jaw based on cone beam computed tomography (CBCT) analysis on the immediate implant placement in this region. Material and Methods:To conduct this research, we used many electronic databases, and the resulting papers were chosen and analyzed. From the clinical point of view, the region of the anterior maxilla is specific and can be difficult for immediate implant placement. Findings: Anatomical structures in the anterior maxilla, such as the nasopalatine canal and accessory canals, may limit and influence the implant therapy outcome. In addition to the aforementioned region, immediate implant placement in the posterior maxilla may be challenging for clinicians, especially in prosthetic-driven immediate implant placement procedures. Data presented within the recently published materials summarize the investigations performed in order to achieve more reliable indicators that may make more accurate decisions for clinicians. Conclusion: The possibility for immediate implant placement may be affected by the NPC shape in the anterior maxilla, while the presence of ACs may increase the incidence of immediate implant placement complications. The variations in IRS characteristics may be considered important criteria for choosing the implant properties required for successful immediate implant placement.

## 1. Introduction

The anatomical border between the anterior and posterior maxilla represents a bony ridge named the zygomaticoalveolar crest. The anterior maxilla is the region that extends from the first premolar on one side to the first premolar on the other side, the cranial border is represented by the anterior nasal opening (the piriform aperture), and the caudal border is the crestal bone [1]. The benefits of cone beam computed tomography (CBCT) usage in planning interventions in the anterior maxilla region have been confirmed in planning implant placement, especially regarding the existence and type of nasopalatine (NPC) and accessory canals (ACs). Also, the importance of CBCT image analyses has been reported in planning interventions in the maxillary molar region. The advantages of such a diagnostic procedure were highlighted in the sense of interradicular septum (IRS) morphological characteristics analysis, which allows for reliable planning of implant therapy.

The use of CBCT imaging in medicine evolved in the 1980s, which was followed by its introduction in dentistry during the 1990s [1]. CBCT is an extraoral radiographic technique that provides a precise 3D image of hard-tissue structures. CBCT imaging is based on the use of a rotating gantry carrying a source of X-ray radiation and a detector. A divergent cone-shaped source of radiation passes directly across the field of interest, and the remaining attenuated radiation is beamed onto the X-ray detector surface on the other side [2,3]. In order to make full volume of the image, only one rotation around the patient is necessary [4]. It is well-known that a significant improvement in the analysis of this maxillary region was achieved by using CBCT. The use of CBCT has overcome the deficiencies of other radiological methods, such as the distortions and superimpositions of 2D radiography, poor resolution, the limited access of computed tomography, the high cost, difficulties in interpretation in dentistry, the longer scanning time, disturbances of metal artifacts, and the high radiation exposure of computed tomography and multi-sliced computed tomography [3]. The use of the CBCT methodology has already been proven to have remarkable characteristics for the subtle morphometric analysis of structures that may be of potential interest in planning the procedures accompanied by immediate implant placement [3,4].

The work’s methodology results in multiplanar views of the dento-maxillofacial region [5]. Also, CBCT provides primary image reconstruction in three fundamental orthogonal planes (Figure 1), such as axial, sagittal, and coronal [6]. Furthermore, the advantages of CBCT analysis provide across-sectional (individual plane) evaluation. It has an importance in the estimation of non-symmetrical structures, such as maxillary sinus septa and alveolar ridge dimensions [7,8]. As the radiation dose in conventional CT and MSCT imaging is up to 10 times higher than in CBCT, CBCT has primacy in the examination of anatomical structures [9]. In addition to the mentioned preference, there are many other advantages, such as the sub-millimeter resolution (2 line pair/mm), images of higher diagnostic quality, shorter scanning times (~60 s), minimal distortion and reduced radiation dosage [2], as well as lower cost than the CT/MSCT [10].

From the clinical point of view, knowledge of anatomical structures and their spatial relationships is crucial in medicine and dentistry for diagnosis and treatment planning [11,12]. Oral and maxillofacial surgery, implantology, orthodontic, ophthalmology, and otorhinolaryngology are fields with increased usage of CBCT [13,14,15,16,17]. The indications for using CBCT have been established by numerous international associations (the Swiss Association of Dentomaxillofacial Radiology and the American Academy of Oral and Maxillofacial Radiology [18,19]). The evaluation of nasopalatine canal (NPC) morphometric characteristics is highly recommended in immediate implant placement planning [20]. Furthermore, CBCT is a reliable radiographic tool used to simulate clinical situations (implant placement) using virtual implants so as to prevent complications [21,22]. Also, virtual planning in maxillofacial surgery is impossible without 3D imaging, and it has been confirmed that CBCT has become a pillar in the design of therapy and in the navigation-based guidance of surgeons during procedures [23,24,25,26]. Furthermore, CBCT scans could allow for navigation implantology, so it is currently considered the gold standard for implant site estimation and treatment planning [27]. The study by Nickenig and collaborators [28] has confirmed that navigation implant placement using CBCT and surgical guides may provide successful implant therapy. It leads to the following advantages: a more precise implantation site, the protection of anatomical structures, high geometrical accuracy, being less invasive, shorter treatment and surgery periods, flapless surgery, and, consequently, less possibility of swelling [29]. Furthermore, 3D bone estimation is recommended for numerous procedures, such as preoperative surgical planning for computed assisted surgeries of tumors and traumas, ridge preservation [30], sinus floor elevation [31,32], and implant placement with bone augmentation [33,34] or phased procedures following block grafts [35,36,37,38].

Therefore, the aim of this narrative review was to highlight recent findings regarding the advantages of using the CBCT methodology in the evaluation of the maxillary region and its specific morphometric characteristics with clinical importance.

## 2. Materials and Methods

Our investigation used the electronic scientific resources PubMed, Scopus, and additional sources, such as Google Scholar and major journals. The search was supplemented with a manual search based on the reference lists of the selected papers and other previous reviews including related journals. Inclusion criteria were studies discussing morphological and morphometric characteristics of NPC, ACs, and IRS using CBCT. Only English-language articles were reviewed. Exclusion criteria included studies with patients under the age of 18. Scientific data were searched during the period between 1 January 2008 and 1 July 2024 to identify primary articles utilizing keywords: “cone beam computed tomography” OR “cone beam CT” OR “CBCT” AND nasopalatine canal OR NPC, accessory canals OR ACs, interradicular septum OR interradicular septal bone OR IRS OR ISB, anterior maxilla OR premaxilla OR alveolar ridge OR posterior maxilla, maxillary central incisors OR MCIs. Those studies that fulfilled the selection criteria were processed for data extraction. After extensive searching, a total of 115 studies were identified and underwent title and abstract screening, and then 61 studies were selected for full-text screening. After full-text screening, 24 studies were excluded. Hence, a total of 37 studies that met our inclusion criteria were processed for data extraction (Figure 2).

## 3. Benefits of CBCT Usage in Planning the Interventions in the Anterior Maxilla Region

Recent studies [13,14] performed using morphometric analyses of CBCT images in the anterior maxilla region performed in order to improve the planning of surgical interventions in that region identified NPC and ACs as potential morphological key structures for the final outcome.

It is well-known that the most prominent anatomical structure in the anterior maxilla is the NPC, the content of which is already described [21,39,40,41,42]. In addition, Mardinger and coworkers [43] classified NPC shapes in four categories, as follows: cylindrical-, funnel-, banana-, and hourglass-shaped.

Distribution of NPC shapes in the study by Mardingler and collaborators [43] was cylindrical, funnel, hourglass, and banana (Figure 3) (50.7, 30.9, 14.5, and 3.9%, respectively). However, Fukuda and coworkers [44] reported the following distribution: funnel, cylindrical, hourglass, and banana (50, 45, 5, and 0%, respectively). Gil-Markques and colleagues [45] noticed that 32.5% of NPCs were banana-shaped, while cylindrical and funnel shapes appeared in 23.5%, while hourglass was the least represented with 20%. In addition, de Lima and coworkers [46] found that the funnel NPC shape was the most frequent, followed by cylindrical, hourglass, and banana NPC shapes (34.1, 27.5, 25.1, and 13.3%, respectively). Recent studies by Milanovic [14] and Arnaut [13] described similar NPC shape distribution. Namely, Milanovic and colleagues [14] noticed that funnel was 35.4%, cylindrical 31%, hourglass 24.8%, and banana 8.8%. Furthermore, Arnaut and coworkers [13] reported that funnel was the dominant NPC shape (34.59%), followed by cylindrical, hourglass, and banana (28.57%, 24.89%, and 12.03%, respectively).

In addition, numerous studies [43,44,45,46,47] examined dimensions of NPC (NPC length, the diameter of incisive foramen, and the nasal foramen diameter). Thus, Mardinger and coworkers [43] noticed that the mean NPC length was 10.7 mm, while the diameters of the incisive foramen and the nasal foramen were 2.94 mm and 2.55 mm, respectively. Etoz and colleagues [47] reported that the mean NPC length and the incisive and the nasal foramen diameter were found at average levels of 12.59 mm, 5.06, and 3.09 mm, respectively. In addition, Fukuda and collaborators [44] quantified the NPC length and the incisive and the nasal diameters as 11.75 mm, 4.28 mm, and 2.84 mm, respectively. Similarly, de Lima and coworkers [46] presented an NPC length of 12.67 mm, while the incisive foramen and the nasal foramen diameters were 3.49 mm and 2.89 mm, respectively.

Furthermore, we analyzed studies that examined the horizontal dimension of the anterior maxilla at different levels that corresponded to postponed NPC [14,39,45]. Thus, Bornstein and colleagues [39] evaluated the horizontal dimension of the anterior maxilla at three consecutive levels. At the lowest level, they observed the minimal dimension of buccal bone wall (6.5 mm) that was followed by the dimension at the middle level (6.59 mm), while the highest thickness (7.6 mm) was observed at the highest level. Gil-Markques and coworkers [45] also measured the horizontal dimension of the anterior maxilla (central incisors region) at three levels and confirmed the same algorithm: the thickness of the buccal bone increased stepwise according to the level of elevation (6.8, 7.2, and 9.9 mm, respectively). On the other hand, Milanovic and collaborators [14] analyzed the horizontal dimension of the anterior maxilla at four consecutive levels (A, B, C, and D) and reported the nonlinear dimensions’ increase (7.11, 7.03, 7.52, and 9.22 mm, respectively). However, only the study by Milanovic and coworkers [14] statistically evaluated the relationship between NPC shape and the horizontal dimension of the anterior maxilla. That methodological approach was used for defining the potential criteria for immediate implant placement in the region of the maxillary central incisors. Severe criteria for expected success are demanded in implant therapy, so it seems necessary to evaluate the morphometric characteristics of the implant site prior to intervention. Hence, Botermans and colleagues [48] recommended the criteria for immediate implant placement in the maxillary esthetic zone. Namely, they proposed that the minimum distances from the labial and palatal bone plates should be 2 mm each. Also, Milanovic and colleagues analyzed the anterior maxilla in the region of the central incisors and gave potential checkpoints in the immediate implant placement planning [14]. Furthermore, the most critical point (insufficient alveolar bone dimension) for the fulfillment of requirements for successful implant therapy was observed in the banana NPC shape. Following the minimal implant diameter (3 mm, [48]) for central incisors immediate implant placement, we present the available bone according to the NPC shape in Table 1.

Other neurovascular structures in the anterior maxilla analyzed for their potential importance in this narrative review were ACs. Numerous studies estimated morphological and morphometric characteristics of ACs [49,50,51,52,53,54,55,56,57]. However, the reported data for the presence of ACs were variable. Thus, Machado [51] reported an incidence of 52.1% and Vasiljevic [49] reported 49.23%, while Baena-Caldas [50] reported that ACs were present in all patients. On the other hand, de Oliveira-Santos [57] observed the presence of ACs in only 15.7%.

In addition, according to ACs’ direction, von Arx and coworkers [52] described three AC shapes (Figure 4) and noticed that the incidence of the curved AC shape was 56.7%, while vertical and Y AC shapes were present in 41.79% and 1.51%, respectively. Tomrukcu and colleagues [53] reported that the curved shape was even more frequent (69.15%) when compared to vertical and Y AC shapes (26.16 and 4.67%). In a study by Vasiljevic and collaborators [49], the prevalence of AC shapes was 48.96%, 36.45%, and 14.58% (curved, vertical, and Y-shaped, respectively).

Numerous studies estimated AC diameters [49,51,53,57] and confirmed that the mean diameter of ACs was approximately 1–1.5 mm. However, the evaluation of the distance between maxillary central incisors and ACs was performed only in the study by Vasiljevic and coworkers [49]. That analysis also provided insight into the relationship between NPC shape and the distance between ACs and maxillary central incisors.

As mentioned above, although not evaluated to the extent of NPC, another anatomical structure that could have an impact on immediate implant placement in the maxillary central incisors region is the AC. The recommended ideal implant position in the anterior maxillary extraction socket is the apico-palatal guiding slot technique (2 mm below the apex of the extraction socket [58]). At the same time, a proposed safety zone of 2 mm from the implant position to the bundle was recommended by Greenstein and colleagues [59]. According to the literature data, it is obvious that ACs’ presence may be considered as a limiting factor for immediate implant placement in the region of maxillary central incisors [59,60,61]. In order to reacha compromise in the recommended data, it is clear that in Table 2 it is quantitatively confirmed that the existence of ACs should be considered as excluding criteria for the implant placement.

Finally, based on all morphometric measurements obtained from CBCT images in numerous studies [13,14,49], provisional guidelines were proposed for the immediate implant placement according to the NPC shape and ACs’ presence (Figure 5).

## 4. Benefits of CBCT Usage in Planning the Interventions in the Maxillary Molar Region

From the aspect of the posterior maxilla, IRS presents the ideal place for implant placement, and it is not surprising that numerous studies estimated morphological IRS characteristics [62,63,64,65,66].

A recent study considering IRS [62] presented morphological and morphometric IRS maxillary molar characteristics at the predefined levels using sagittal and axial CBCT slices. There were no significant differences (sagittal slices) reported between IRS characteristics of the maxillary molars (both) in bilateral comparison. The same confirmation was noticed using axial CBCT slices. In addition, differences between IRS characteristics of the first and second maxillary molars were evaluated. Namely, using sagittal CBCT slices, we concluded that the interradicular septum width of the first molars was significantly higher when compared to the second maxillary molars at all evaluated IRS height levels. The same results were reported for the interradicular furcation angle. On the other hand, the IRS height of the second molars was significantly higher than the IRS of the first molars (7 mm and 6.5 mm). Agostinelli and coworkers [66] reported a higher value for the IRS height of the first maxillary molars compared to the second (4.51 vs. 3.24 mm). Finally, no significant difference was shown in the distance between the IRS base and the sinus floor of the first and second molars.

Furthermore, using axial CBCT slices, the perimeter and IRS surface area of the first and second molars were estimated [63]. The average first maxillary molars’ perimeter and surface area values were app. 11 mm and 5.5 mm^2^ at the lowest level, with stepwise increases in perimeter (12.5, 14, and 16 mm) and surface area (8, 9, and 12.5 mm^2^) in the higher sectors. The IRS perimeter and surface area of the second maxillary molars were significantly lower when compared to the first molars. Again, from the bottom to the top, a stepwise increase was observed both for the perimeter (7.5, 10, 11, and 13 mm) and the surface area (3, 4.5, 5.5. and 6.5 mm^2^) of the second maxillary molars. A similar observation was reported by Agostinelli and colleagues [66].

Because a possible relationship between different anatomical structures was noticed in previous studies [13,14], the interconnection between the IR furcation angle and the IRS surface area/height at level A for both maxillary molars was estimated. Namely, the correlation between the IR furcation angle and the IRS surface area at level A was significantly positive, while it revealed a significant negative correlation between the IR furcation angle and the IRS height for both groups of molars [63].

This was soon followed by a recent study [63], which proposed the classification of the IRS structure into five different shapes (arrow-, boat-, drop-, palatal-convergence-, and buccal-convergence-shaped) based on coronal CBCT slice analysis (Figure 6). This classification resulted in predefined quantitative criteria for maxillary molars’ IRS. The maxillary molars’ IRS widths at the base level for the arrow, boat, palatal, and buccal convergence were ≥4 mm. In contrast, for the drop IRS shape, the IRS width was ≤4 mm, which was also the case for both maxillary molars. The second (and final) principal criterion for IRS shape classification was the IR furcation angle. The first molars’ IR furcation angles for the arrow and palatal convergence shapes were ≤60° and ≤70°, respectively. On the other hand, the IR furcation angle of buccal convergence, drop, and boat IRS shapes were ≥60°, ≥70°, and ≥90°, respectively. Less variability in the IR furcation angle was shown in the second molars. Hence, the arrow, drop, palatal, and buccal convergence shapes presented IR furcation angles ≤70°, while in the boat IRS shape the angle was ≥70°. It was also noticed that the most frequent IRS shape was arrow (app. 45%), while the lowest frequency was observed for the drop-shaped IRS (app. 11.5%). Similar variability in IRS shapes’ appearance was present in both the first and the second maxillary molars.

In the same study [63], the IRS shape significantly affected all examined parameters (IRS width at the different levels, IR furcation angle, IRS height, distance between IRS base and sinus floor, distance between IR furcation and sinus floor, as well as IRS surface area at all estimated IRS levels). For the first maxillary molars, at all evaluated IRS levels obtained on the coronal view, palatal convergence showed the highest value for the IRS width, as well as height. Afterwards, the IR furcation angle expressed the highest values in the boat shape. This shape also expressed the smallest distance between the IR furcation and the sinus floor. For the second molars, the drop-shaped IRS presented the narrowest IRS at all estimated levels, but with the highest septum. The same was true for the first molars;the largest angle and the smallest distance between the IR furcation and the sinus floor were in the boatshape for the second molars.

Finally, the results obtained on axial slices [63] confirmed that the largest IRS surface area (at all estimated levels) was in the palatal convergence IRS shape for the first upper molars group. Interestingly, the results of the same study showed that the variations in the IRS surface area for the second molars were more prominent, as the highest values for the IRS surface area were observed at the different IRS heights for various IRS shapes.

Also, several authors gave recommendations for IRS dimensions (minimum width of 3 mm and height of 10 mm) for successful immediate implant placement [67,68]. According to this recommendation, we give a suggestion for immediate implant placement based on IRS shape morphometric characteristics [63] in Figure 7. On the other hand, some interventions, such as ossedensification and crestal sinus lift, can overcome this limitation. Namely, both IRS ossedensification and crestal sinus lift elevation can be performed in one-stage surgery. Numerous studies confirmed high rates of success of this intervention [69,70].

In addition, following the even more rigorous criterion, based on the fact that the most commonly used implant diameter in the posterior maxilla is 4 mm [69,70], in Figure 8, an overview is presented confirming that none of IRS shapes provides sufficient space for reliable implant placement stability. The proposed analysis is in line with the previous report that described difficulties with immediate implant placement into IRS [71].

## 5. Discussion

As it is already known, CBCT provides precise bone measurements in order to prevent complications during interventions in the anterior maxilla [13,14]. One of the main criteria for successful immediate implant placement in the region of maxillary central incisors is sufficient thickness of the labial alveolar bone [72]. Thus, some authors [73,74] reported that labial alveolar bone thickness should be at least 2 mm to avoid labial gingival recession and so that biological and esthetic outcomes are achieved. In addition, CBCT examination of the alveolar ridge inclination during implant planning therapy is highly recommended [20]. Namely, immediate implant placement in the region of the anterior maxilla, especially in the lateral incisors area, could be compromised by facial bone concavity [75,76,77].

On the other hand, it has been confirmed in recent studies that anatomical structures in the anterior maxilla, such as the NPC and accessory canals (ACs), could also compromise interventions in that region [21,49]. It is not surprising that numerous studies performed using CBCT images’ morphometric analyses in the anterior maxilla region with the aim to improve the planning of surgical interventions in that region pointed to the NPC and ACs as potential morphological key structures for the final outcome [13,14].

Several NPC morphological classifications have been proposed in the current literature data [43,47,78,79].Milanovic and collaborators evaluated the dimensions of the alveolar bone (as the common place for NPC and AC appearance) andthe thickness at the four consecutive height levels and concluded that the inferior parts of the anterior maxilla presented a lesser dimension than the superior parts [14]. Their findings were in line with the previous observations by Güncü [80] and Lopez and collaborators [81]. A recent study [14] estimated the impact of NPC shape on the anterior maxilla dimensions and confirmed that the impact existed. Namely, except at the lowest level, patients with the banana NPC shape had the smallest horizontal dimension of the anterior maxilla [14].

In order to achieve adequate morphometric conditions for immediate implant placement, especially in patients with a banana NPC shape, several additional surgical interventions (such as bone augmentation and NPC exploration) have been proposed [82,83]. Also, using the NPC as an implant place can be considered [84]. Furthermore, it was suggested by Singhal and coworkers [85] to use surgical guides for precise implant placement into nasopalatine foramen. In patients with funnel NPC shapes, simultaneous use of surgical guides during implant insertion has been recommended [86].

In contrast to the commonly accepted opinion that the NPC is the only neurovascular structure in the anterior maxilla with clinical importance [87], the importance of the estimation of the additional existing channels was highlighted in recent studies [49,50]. A recent study [49] confirmed that the ACs were present in the region of the maxillary central incisors in 49.23% of patients, which is in line with von Arx and colleagues [52], while Wanzeler and collaborators found ACs in almost 90% [54]. Bilateral localization was reported as the most frequent in estimation of ACs’ localization by Anatoly and coworkers [55], which is in accordance with our previous study [49]. On the other hand, Manhães and collaborators showed that the variations in AC distribution were side-dependent [56]. Machado [51] and Vasiljevic and coworkers [49] also evaluated ACs’ diameter and showed that it was approximately 1 mm, while Oliveira Santos and colleagues [57] reported a slightly larger AC diameter.

Furthermore, there are numerous studies that assessed the interconnection between NPCs and ACs and related anatomical structures [14,49,88]. In addition, the specific relationship between those two vulnerable anatomical structures (NPCs and ACs) in anterior maxilla was estimated by Vasiljevic and coworkers [49], confirming that the described relationship significantly depends on the NPC shape.

It has been confirmed that the NPC’s and ACs’ localization could be a limiting factor for dental implant placement [14,89]. The orthognathic surgical interventions, such as Le Fort 1, require examination of anatomical structures and careful treatment planning to avoid neurovascular complications [90]. From the point of view of implantology, Botermans and coworkers reported that a secure zone between the NPC and the implant is required [48]. However, several virtual implant placement studies showed that it was difficult to achieve those recommendations [21,91]. Furthermore, Botermans and collaborators [48] noticed NPC perforation during virtual immediate implant placement in about 53% of cases, while Jia and colleagues [91] showed slightly above 16%. Also, the damage toACs during implant placement is described in the literature [89]. It is confirmed that the damage to anatomical structures in the anterior maxilla could be followed by several complications, such as non-integration of a dental implant, mucosa necrosis, pain, paresthesia, hemorrhage, the sensation of the burning head in the occipital region, and neuropathy [89,92].

Summarizing the algorithm presented in Figure 5, it is obvious that the patients with a banana NPC shape represent the most vulnerable group for immediate implant placement. On the other hand, the morphometric characteristics of the anterior maxilla in subjects with a cylindrical NPC shape are considered to require additional interventions (such as bone grafting) in order to qualify for immediate implant placement. Hourglass and funnel shapes of the NPC, according to our previous results, allow sufficient operating space for surgical interventions. In order to achieve adequate morphometric conditions for the immediate implant placement, especially in patients with a banana NPC shape, several additional surgical interventions (such as bone augmentation and NPC exploration) have been proposed [82,83]. Also, it could be considered to use NPC as an implant place [83,84]. Furthermore, it was suggested by Singhal and coworkers [85] to use surgical guides for precise implant placement into nasopalatine foramen. In patients with a funnel NPC shape, simultaneous use of surgical guides during implant insertion has been recommended [86].

As seen in Figure 5, the presence of ACs should be considered as an absolute contraindication for immediate implant placement. So, the clinicians should follow Shelley’s [93] recommendation that in those cases there is a need for other therapeutic options, such as a realistic fixed alternative (adhesive cantilever bridge), in order to avoid potentially damaging implant therapy

Because the posterior maxillary parts are often exposed to several [94,95,96,97,98,99] surgical interventions (such as implant placement therapy, Le Fort I osteotomy, sinus surgery and sinus lift elevation, cyst enucleation, tumor extirpation, surgical tooth extraction, and present donor site area), the accurate morphometric analysis of CBCT images is required for reliable planning.

The anatomical structures in posterior maxillary parts show high variability [100,101,102]. From the clinical point of view, IRS has been frequently evaluated as the ideal place for implants [63,103]. Numerous advantages of implant placement in maxillary molars IRS are described, such as the possibility of achieving primary implant stability, adequate force distribution, and plaque control [104,105]. On the other hand, anatomical challenges, as well as the differences between implant diameter and post-extractive alveoli, quality and quantity of bone, root length and configuration, the root trunk height, and roots’ divergence, could make surgical interventions more difficult [67,106,107,108,109]. Several IRS classifications according to the different criteria have been offered in the literature [63,64]. Namely, Bleyan and collaborators [64] described four categories based on the IRS width (after tooth extraction), as follows: S-I—septum initial width above 4 mm; S-II—septum initial width 3–4 mm; S-III—septum initial width 2–3 mm; and S-IV—septum initial width below 2 mm or no septal bone present. On the other hand, Milenkovic and coworkers [63] defined five IRS shapes based on linear measurements and visual impressions (arrow-, boat-, drop-, palatal-convergence-, and buccal-convergence-shaped) by analyzing CBCT images. However, Smith and colleagues [65] estimated the relationship between the implant and the IRS and noticed three categories: type A sockets (alveoli have sufficient septal bone, which circumferentially surrounds the coronal implant part), type B sockets (the septal bone does not cover the total implant surface, but it is enough to achieve primary implant stability), and the least, type C (has an insufficient septal bone to stabilize the implant without engaging the socket wall).

In post-extractive alveoli, clinicians could also use one of the root cavities to insert the implant in order to achieve primary implant fixation, but it does not meet the requirements for the correct prosthetic implant position [110]. However, previous studies reported that IRS presents the ideal implant site from a prosthetic view [111]. It is not surprising that Rajkovic Pavlovic and coworkers [112] emphasized the importance of radiological IRS evaluation in implant placement planning due to the fact that primary implant stability is determined by the alveoli architecture [113]. In order to obtain better insights into IRS morphological characteristics, Milenkovic and coworkers [63] presented criteria for the classification of five IRS shapes. As presented in Figure 7, the minimal clinical margin for achieving the primary implant stability could not be established only for the drop-shaped IRS, according to the criterion of IRS width. In contrast, this indicator for the primary implant stability in the second molars was achieved only in the boat IRS shape. According to the implant height, as the criterion necessary to achieve implant stability, none of the IRS shapes allows for successful implant placement.

Although previous analyses do not look optimistic from the point of implant placement success, it should be noted that clinicians have a plethora of possibilities to achieve the final success of implant stability by applying additional interventions. Those interventions may be performed prior to implant placement, simultaneously, and/or following the standard implant placement procedure. Some of them are aimed at increasing the IRS width, while the others are recommended for the enlargement of the IRS height. As previously commented, both goals have to be achieved. However, the final success of those procedures accompanied with the implant placement is still under evaluation.

Considering the procedures for the enhancement of IRS width, it should be noticed that osseodensification is a novel method of biomechanical bone preparation in order to achieve improvement in better implant stability when compared to the standard drill [69]. Hence, in the clinical study conducted by Bleyan and coworkers [64], theosseodensification method to improve IRS dimensions was used. They reported the success of implant therapy by means of a 93.1% survival rate at the 12-month follow-up (135/145 implants). Another surgical technique that improves the post-extractive alveoli volume is socket grafting [114,115,116,117]. There are numerous materials for filling the gap between implants and alveoli described in the literature [118,119,120]. The clinical studies confirmed the success of implant therapy by substituting the insufficient implant sites [65,121].

In cases with insufficient bone height, the sinus lift procedure aims to improve vertical bone dimensions below the maxillary sinus in order to provide the possibility of implant placement [122]. Also, the sinus membrane perforation risk could be reduced through this procedure while simultaneously achieving primary implant stability [123]. On the other hand, the dimensions of membrane perforation play an important role in implant placement success [124,125] Thus, Lombardo and coworkers concluded that internal sinus lift presented a small risk of membrane perforation and stable bone around implants with a high rate of success at mid-term follow-up for atrophic maxilla [126].

The mentioned procedures aimed to enhance IRS width and IRS height, and in relation to the analysis presented in the study that analyzed IRS shapes, it is worth noting that CBCT images’ morphometric analysis allows for the achievement of exact morphometric parameters that can lead to better planning of interventions in the maxillary molar region.

On the other hand, modern implant therapy includesshort implants with wide threads, such as AnyRidge implants (Anyridge^®^, Megagen, South Korea) [127]. Numerous studies concluded that using this implant type may achieve high primary stability in immediate implant placement for the maxillary posterior region [128,129].

## 6. Conclusions

Taken altogether, the impact of the methodological approach based on CBCT image analysis of maxilla, as presented in this summary of our recent studies, is manifested by achieving more reliable recommendations for performing certain interventions in the maxillary region. Summarizing the proposed operating procedures, according to our recent investigations based on the potential usefulness of CBCT-based diagnostic procedures in planning some specific interventions in the described regions of interest, it seems that the following can be stated:The possibility for immediate implant placement may be affected by the NPC shape in the anterior maxilla, while the presence of ACs, in general, may increase the incidence of immediate implant placement complications;The variations in IRS characteristics may be considered important criteria for choosing the implant properties required for successful immediate implant placement.

The presented relationships of potential clinical importance may be considered as tentative factors for exclusion criteria under the described circumstances. Furthermore, in the case that there are no absolute contraindications for the interventions in this region, the use of the proposed methodology allows for better intervention planning by means of predefining optimal procedures and quantification of the extent of accompanied treatments.

## 7. Future Directions

The recent literature provides exact data on the anatomical configuration of the upper jaw that may affect implant placement. However, subsequent studies could be directed towards virtual implant placement in order to obtain more precise insight into the relationship between the implant and vulnerable anatomical structures, such as the NPC, ACs, and maxillary sinus membrane. Beside gold standards for implant macrodesign, using virtual planning software for implant surgery with a different macrodesign should also be examined, as well as the relationship between such a design and the aforementioned anatomical structure. 

## Figures and Tables

**Figure 1 diagnostics-14-01697-f001:**
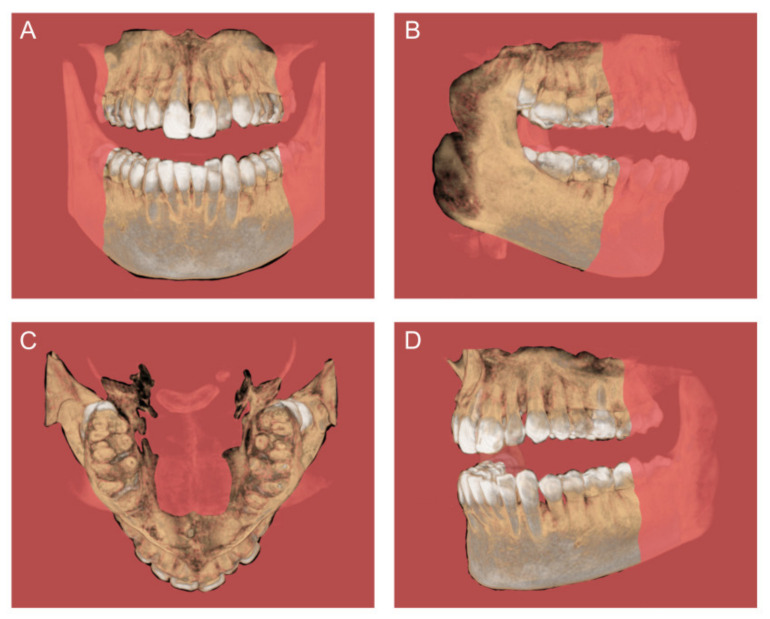
CBCT images with the orthogonal planes: (**A**) coronal, (**B**) sagittal, (**C**) axial, and (**D**) individual plane.

**Figure 2 diagnostics-14-01697-f002:**
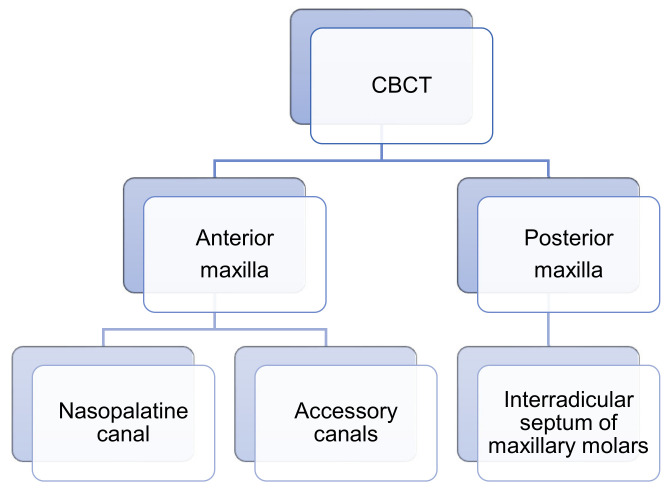
The scheme of the analyzed anatomical structures of the anterior and posterior maxilla through CBCT.

**Figure 3 diagnostics-14-01697-f003:**
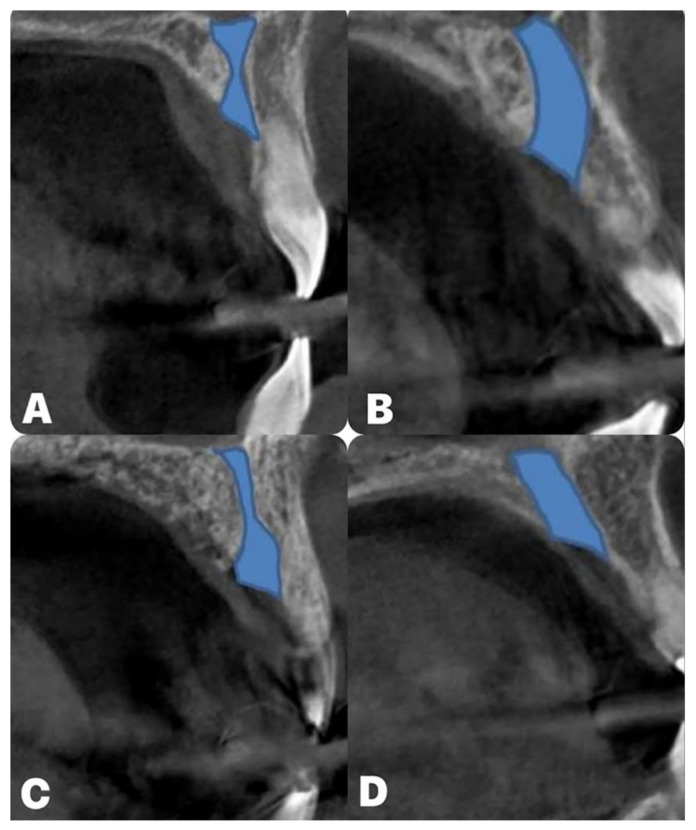
The different NPC types: (**A**) hourglass; (**B**) banana; (**C**) funnel; (**D**) cylindrical.

**Figure 4 diagnostics-14-01697-f004:**
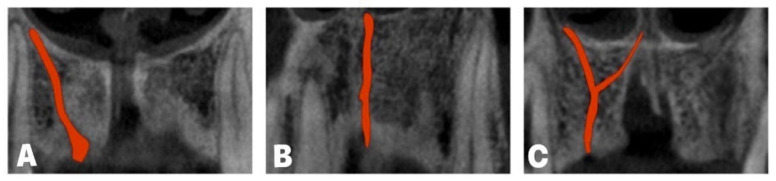
Accessory canal shapes: (**A**) curved; (**B**) vertical; (**C**) Y-shape.

**Figure 5 diagnostics-14-01697-f005:**
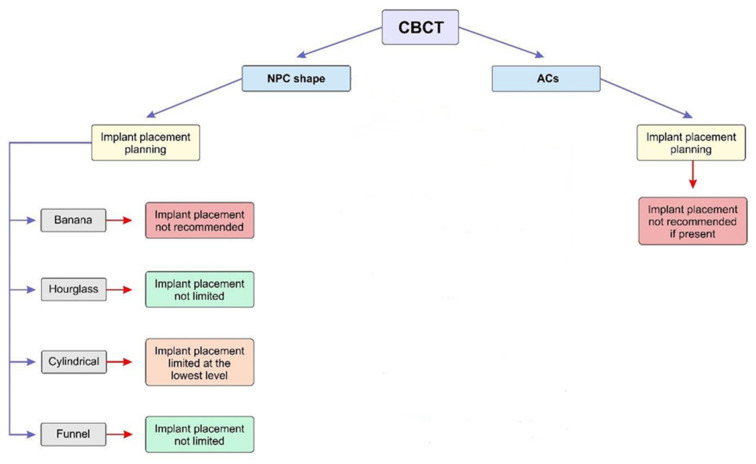
Proposed recommendations for implant placement according to the NPC shape and ACs present. Recommendations for implant placement are based only on the morphometric characteristics without additional interventions (ossedensificaton, bone grafting, and NPC exploration).

**Figure 6 diagnostics-14-01697-f006:**
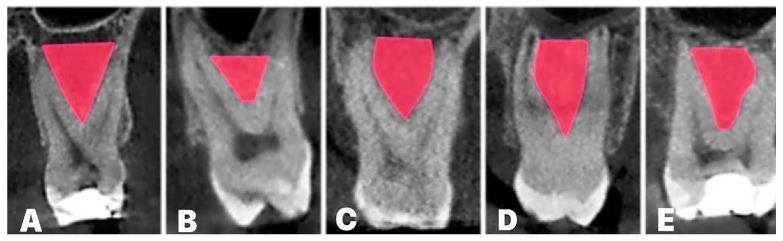
Maxillary molars’ IRS shapes:(**A**) arrow; (**B**) boat; (**C**) drop; (**D**) palatal convergence; (**E**) buccal convergence.

**Figure 7 diagnostics-14-01697-f007:**
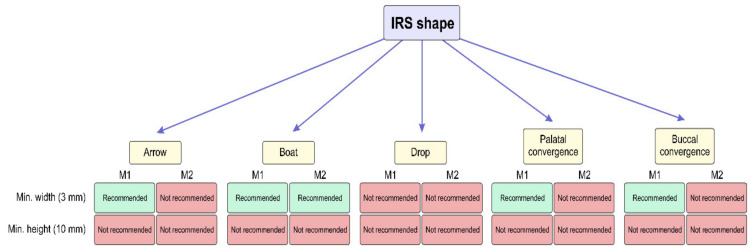
Proposed implant placement recommendations according to the morphometric characteristics depending on IRS shape (without additional surgical treatment).

**Figure 8 diagnostics-14-01697-f008:**
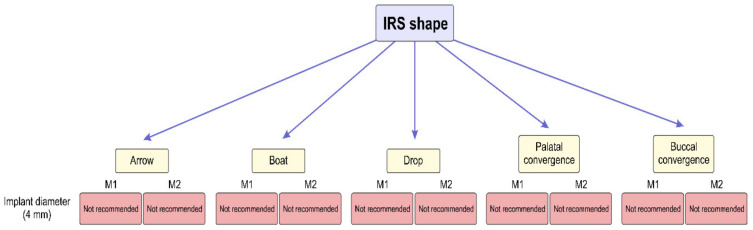
The analysis of IRS morphometric potentials required for reliable implant placement stability according to the IRS shape. (Proposed recommendations for the implant placement are based only on the IRS morphometric characteristics without additional surgical treatment).

**Table 1 diagnostics-14-01697-t001:** Available bone dimension for the implant placement in the anterior maxilla according to the NPC shape (in mm). Insufficient space (marked in red); minimal required dimensions (marked in yellow); sufficient space (marked in green).

	NPC Shape			
The Horizontal Dimensions of the Anterior Maxilla at Different Levels	Banana	Hourglass	Cylindrical	Funnel
Level A	2.3	3.2	2.9	3.5
Level B	1.4	3.4	3.2	3
Level C	1.6	4	3.6	3.7
Level D	2	5.9	5.7	5.8

**Table 2 diagnostics-14-01697-t002:** Average palatal distance between accessory canals (ACs) and maxillary central incisors (MCIs) for implant placement according to NPC shape (in mm). Insufficient distance marked in red.

	NPC Shape			
ACs–MCIs Distance at the Different Levels	Banana	Hourglass	Cylindrical	Funnel
Level A	1.18	0.9	0.8	1.05
Level B	1.19	1.1	1.08	1.17
Level C	0.8	1.25	1.7	1.3

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
