# Peer review of "Anatomical Factors of the Anterior and Posterior Maxilla Affecting Immediate Implant Placement Based on Cone Beam Computed Tomography Analysis: A Narrative Review"

_diagnostics, 2024, doi:10.3390/diagnostics14151697_

Round 1

Reviewer 1 Report

Comments and Suggestions for Authors

Dear Authors,

In the following text, find brief indicators of my remarks for improving the submitted version of the manuscript titled "Anatomical Factors of the Anterior and Posterior Maxilla Affecting Immediate Implant Placement: A Narrative Review" (manuscript ID: diagnostics-3131076): 

1. In the abstract of the manuscript, please clearly state the aim of the review and the conclusion of the review.

2. The main text of the manuscript would significantly benefit from the addition of a robust literature search strategy. This will not only enhance the credibility of the review but also guide the readers on the selection process of the included and excluded papers.

3. Figure 1: The figure caption is unsuitable for the figure content. The advantages and clinical importance were not indicated in the figure, so please revise the content of the figure to match the figure caption. Also, a graphic presentation of CBCT findings with clearly marked anatomical structures would be an interesting addition to the manuscript.

4. A graphical representation (either CBCT or schematic) of the shapes of nasopalatine canals and accessory canals might be beneficial for readers who are not familiar with the topic.

5. Figure 3: The figure is not nicely placed on the page, so it could not be reviewed fully. Please correct it.

6. Figure 4: Please increase the font size for the smallest text because it is tough to read.

7: It would be better if the authors consider removing the discussion as a separate paragraph and including these points in the previous chapters since most of the information is repeated.

8. Please carefully correct the abbreviations and formatting of the main text.

More detailed comments and concerns are given in the attached PDF file, so please look at it carefully to revise the manuscript.

Comments on the Quality of English Language

Minor English editing required.

Author Response

Dear Reviewer, thank you for your valuable suggestions to improve our manuscript. We have made the following changes in response to your comments:

Comments 1: In the abstract of the manuscript, please clearly state the aim of the review and the conclusion of the review.

Response 1: As you suggested, we have clearly stated the aim and conclusion of the review in the abstract:

“The aim of this narrative review was to provide insights into the influence of the morphological characteristics of the anatomical structures of the upper jaw based on cone beam computed tomography (CBCT) analysis on the immediate implant placement in this region. The possibility for immediate implant placement may be affected by NPC shape in anterior maxilla, while the presence of ACs may increase the incidence of immediate implant placement complications. The variations of IRS characteristics may be considered as important criteria for choosing the implant properties required for successful immediate implant placement.”

Comments 2: The main text of the manuscript would significantly benefit from the addition of a robust literature search strategy. This will not only enhance the credibility of the review but also guide the readers on the selection process of the included and excluded papers.

Response 2: Following your suggestions, we added search strategy for our manuscript. Our investigation used the electronic scientific resources PubMed, Scopus and additional sources, such as Google Scholar and major journals. The search was supplemented with a manual search based on the reference lists of the selected papers and other previous reviews including related journals. Inclusion criteria were studies discussing morphological and morphometric characteristics of NPC, ACs, IRS using CBCT. Only English language articles were reviewed. Exclusion criteria included studies with patients under the age of 18. Scientific data were searched in the period between January 1, 2008, to July 1, 2024, to identify primary articles utilizing keywords: “cone beam computed tomography” OR “cone beam CT” OR “CBCT” AND nasopalatine canal OR NPC, accessory canals OR ACs, interradicular septum OR interradicular septal bone OR IRS OR ISB, anterior maxilla OR premaxilla OR alveolar ridge OR posterior maxilla, maxillary central incisors OR MCIs. Those studies that fulfilled the selection criteria were processed for data extraction. After extensive searching, a total of 115 studies were identified and underwent title and abstract screening, then 61 studies were selected for full text screening. After full text screening, 24 studies were excluded. Hence, a total of 37 studies that met our inclusion criteria were processed for data extraction.

Comments 3: Figure 1: The figure caption is unsuitable for the figure content. The advantages and clinical importance were not indicated in the figure, so please revise the content of the figure to match the figure caption. Also, a graphic presentation of CBCT findings with clearly marked anatomical structures would be an interesting addition to the manuscript.

Response 3: We have replaced Figure 1 in Material and Methods section and modified the caption. Namely, this figure is important for our manuscript, because this is a schema which represents examined anatomical structures in this Narrative review. Also, we added a graphic presentation of CBCT findings with clearly marked anatomical structures.

Comments 4: A graphical representation (either CBCT or schematic) of the shapes of nasopalatine canals and accessory canals might be beneficial for readers who are not familiar with the topic.

Response 4: In response to your suggestion, we have added CBCT images showing the shapes of the nasopalatine canals, accessory canals, and interradicular septum to aid readers who may not be familiar with these structures.

Comments 5: Figure 3: The figure is not nicely placed on the page, so it could not be reviewed fully. Please correct it.

Response 5: The placement of Figure 3 has been corrected to ensure it is fully visible.

Comments 6: Figure 4: Please increase the font size for the smallest text because it is tough to read.

Response 6: We have increased the font size in Figure 4 to improve readability.

Comments 7: It would be better if the authors consider removing the discussion as a separate paragraph and including these points in the previous chapters since most of the information is repeated.

Response 7: We have considered your suggestion and removed repeated content from the Discussion, now we focus on providing a detailed explanation of the relationship between anatomical structures and immediate implant placement, also from a clinical point of view. On the other hand, many recommendations about surgical techniques such as guided surgery in critical cases (insufficient bone due to anatomical factors), were mentioned with detailed information in discussion section. Similarly, issues related to the presence of ACs in patients requiring implant placement are discussed in detail only in the Discussion section. All information from section 3 and 4covered in greater detail in the Discussion section. We hope these revisions make the manuscript more acceptable.

Comments 8: Please carefully correct the abbreviations and formatting of the main text.

Response 8: We have corrected the abbreviations and added a list of abbreviations. Additionally, we have formatted the main text according to your suggestions.

Comments 9: Minor English editing required

Response 9: Minor English language corrections have been made as recommended.

More detailed comments and concerns are given in the attached PDF file, so please look at it carefully to revise the manuscript.

We have addressed all your suggestions, including descriptions of anterior and posterior maxilla borders, correction of typos, addition of new references, rephrasing of unclear sentences etc.

Thank you once again for your thorough review and constructive feedback.

Reviewer 2 Report

Comments and Suggestions for Authors

Dear authors,
thank you for the opportunity to revise this manuscript. The topic is of great interest.

Please revise the following aspects.

-Title: As “the aim of this narrative review was to highlight recent findings regarding the advantages of using CBCT methodology in the evaluation of maxillary region and its specific morphometric characteristics with clinical importance”, I suggest to change the title focusing on the use of CBCT.

-Abstract: Please divide in subsections.

-Introduction: Please add comparison between details visible with CBCT and info visible with other types of radiograph, for implant planning.

-paragraph 2: Figure 3 is cut, please provide it entirely.

-paragraph 3: “Accompanying instructions for successful immediate implant placement such as IRS minimum width of 3 mm [67] and minimum height of 10 mm [68], are proposed as recommendations in Fig. 4”: please better discuss this point also regarding the possibilities for therapy implant planning.

-Discussion: “in cases with insufficient bone height, the sinus lift procedure aims to improve vertical bone dimension below the maxillary sinus in order to provide the possibility of implant placement [122]. Also, sinus membrane perforation risk could be reduced by this procedure simultaneously with achieving primary implant stability [123]”: several authors described this issue, please add some content at this regard, considering also mini-invasive techniques available (see e.g. 10.1186/s40729-021-00346-7, 10.1111/prd.12286, 10.3390/ma15227995).

-Please add a list of abbreviations.

-Please add limitations to the review design, which is narrative and not systematic. Why did you choose it?

Comments on the Quality of English Language

Moderate English checking is required, minor grammar mistakes.

Author Response

Dear reviewer, thank you for your valuable suggestions to improve our manuscript. We have addressed your comments as follows:

Comments 1: Title 1: As “the aim of this narrative review was to highlight recent findings regarding the advantages of using CBCT methodology in the evaluation of maxillary region and its specific morphometric characteristics with clinical importance”, I suggest to change the title focusing on the use of CBCT.

Response 1: Based on your suggestion to focus on CBCT usage, we have revised the title to: “Anatomical Factors of the Anterior and Posterior Maxilla Affecting Immediate Implant Placement Based on CBCT Analysis: A Narrative Review.”

Comments 2: Abstract: Please divide in subsections.

Response 2: We have divided the Abstract into the following subsections: Background, Material and Methods, Findings, and Conclusion

Comments 3: Introduction: Please add comparison between details visible with CBCT and info visible with other types of radiograph, for implant planning.

Response 3: We have added a comparison between the details visible with CBCT and other types of radiographs for implant planning. The improved section now reads: "The use of CBCT has overcome the deficiencies of other radiological methods such as the distortions and superimpositions of 2D radiography, poor resolution, limited access of computed tomography, high cost, difficulties in interpretation in dentistry, longer scanning time, disturbance of metal artifacts, and high radiation exposure of computed tomography and multi-sliced computed tomography [3]. The use of CBCT methodology has already proven remarkable characteristics for the subtle morphometric analysis of structures that may be of potential interest in planning the procedures accompanied by immediate implant placement [3,4]."

Comments 4: -paragraph 2: Figure 3 is cut, please provide it entirely.

Response 4: Figure 3 has been corrected and is now fully visible.

Comments 5: -paragraph 3: “Accompanying instructions for successful immediate implant placement such as IRS minimum width of 3 mm [67] and minimum height of 10 mm [68], are proposed as recommendations in Fig. 4”: please better discuss this point also regarding the possibilities for therapy implant planning.

Response 5: We have expanded the discussion regarding the recommendations for IRS dimensions and their implications for therapy and implant planning. The revised section now reads: "Also, several authors gave recommendations for IRS dimensions (minimum width of 3 mm, and height 10 mm) for successfull immediate implant placement [67, 68]. According to this recommendation we give a suggestion for immediate implant placement based on IRS shape morphometric characteristics [63] in Fig. 7. On the other hand, some interventions such as ossedensification and crestal sinus lift can overcome this limitation. Namely, both IRS ossedensification and crestal sinus lift elevation can be performed in one-stage surgery. Numerous studies confirmed high rates of success of this intervention [69,70].”

Comments 6 :-Discussion: “in cases with insufficient bone height, the sinus lift procedure aims to improve vertical bone dimension below the maxillary sinus in order to provide the possibility of implant placement [122]. Also, sinus membrane perforation risk could be reduced by this procedure simultaneously with achieving primary implant stability [123]”: several authors described this issue, please add some content at this regard, considering also mini-invasive techniques available (see e.g. 10.1186/s40729-021-00346-7, 10.1111/prd.12286, 10.3390/ma15227995).

Response 6: We have revised this section to include more information from the literature and address mini-invasive techniques based on references as you suggested. The updated content now reads: “In cases with insufficient bone height, the sinus lift procedure aims to improve vertical bone dimension below the maxillary sinus in order to provide the possibility of implant placement [122]. Also, sinus membrane perforation risk could be reduced by this procedure simultaneously with achieving primary implant stability [123]. On the other hand, dimensions of membrane perforation play an important role in implant placement success [124,125] Thus, Lombardo and coworkers concluded that internal sinus lift presented small risk for membrane perforation and stable bone around implants with high rate success at mid-term follow-up for atrophic maxilla [126].”

Comments 7: Please add a list of abbreviations.

Response 7: Following your instructions , we have added the list of abbreviations: CBCT – Cone Beam Computed Tomography, NPC – Nasopalatine Canal, ACs – Accessory Canals, IRS – Interradicular Septum.

Comments 8: Please add limitations to the review design, which is narrative and not systematic. Why did you choose it?

Response 8: According to your comment we added limitations of the review design and explain why we have chosen a narrative type of review. Due to limited number of studies on this subject and considering that this is one of the first reviews in recent literature based on clinical importance and direct connection between mentioned anatomical structures and immediate implant placement we decided on a narrative review.  We hope that in the future there will be a greater number of studies based on a similar methodology that would enable the creation of a systematic review article. In this article the emphasis is on recommendations that may help clinicians in implant placement planning, and not on statistical reports of the literature.

Comments 9: Moderate English checking is required, minor grammar mistakes.

Response 9: English language corrections have been made as advised.

Round 2

Reviewer 1 Report

Comments and Suggestions for Authors

Dear Authors,

thank you for addressing my previous comments and concerns. I have no further suggestions related to this manuscript. 

Comments on the Quality of English Language

Minor English editing required.